# Tissue-resident macrophages promote extracellular matrix homeostasis in the mammary gland stroma of nulliparous mice

Ying Wang[1†], Thomas S Chaffee[1†], Rebecca S LaRue[2], Danielle N Huggins[1], Patrice M Witschen[3], Ayman M Ibrahim[4,5], Andrew C Nelson[1,6], Heather L Machado[4], Kathryn L Schwertfeger[1,6,7]*

[1]Department of Laboratory Medicine and Pathology, University of Minnesota, Minneapolis, United States; [2]University of Minnesota Supercomputing Institute, University of Minnesota, Minneapolis, United States; [3]Comparative and Molecular Biosciences Graduate Program, University of Minnesota, Minneapolis, United States; [4]Department of Biochemistry and Molecular Biology, Tulane Cancer Center, Tulane School of Medicine, New Orleans, United States; [5]Department of Zoology, Faculty of Science, Cairo University, Giza, Egypt; [6]Masonic Cancer Center, University of Minnesota, Minneapolis, United States; [7]Center for Immunology, University of Minnesota, Minneapolis, United States

**\*For correspondence:**
schwe251@umn.edu

[†]These authors contributed equally to this work

**Competing interests:** The authors declare that no competing interests exist.

**Abstract** Tissue-resident macrophages in the mammary gland are found in close association with epithelial structures and within the adipose stroma, and are important for mammary gland development and tissue homeostasis. Macrophages have been linked to ductal development in the virgin mammary gland, but less is known regarding the effects of macrophages on the adipose stroma. Using transcriptional profiling and single-cell RNA sequencing approaches, we identify a distinct resident stromal macrophage subpopulation within the mouse nulliparous mammary gland that is characterized by the expression of Lyve-1, a receptor for the extracellular matrix (ECM) component hyaluronan. This subpopulation is enriched in genes associated with ECM remodeling and is specifically associated with hyaluronan-rich regions within the adipose stroma and fibrous capsule of the virgin mammary gland. Furthermore, macrophage depletion leads to enhanced accumulation of hyaluronan-associated ECM in the adipose-associated stroma, indicating that resident macrophages are important for maintaining homeostasis within the nulliparous mammary gland stroma.

## Introduction

Tissue-resident macrophages have been ascribed various functions, including immune surveillance and phagocytosis of apoptotic debris (*Davies et al., 2013*). Depending upon the site of localization, these macrophages can exhibit additional tissue-specific behaviors, such as the regulation of surfactant by alveolar macrophages in the lung and of neuronal function by microglia in the brain (*Haldar et al., 2014*). These tissue-specific functions are thought to be driven in response to factors within the local tissue microenvironment. Tissue-resident macrophages are typically derived from embryonic precursors, and can maintain locally through self-renewal or be replaced by blood-derived monocytes, or a combination of both, depending on the tissue (*Ginhoux and Guilliams, 2016*; *Hashimoto et al., 2013*; *Perdiguero et al., 2015*; *Sheng et al., 2015*; *Yona et al., 2013*; *Zhu et al., 2017*). Recent studies of macrophage ontogeny and localization have highlighted the

complex nature of macrophage heterogeneity and function at steady state (*Chakarov et al., 2019*). Further understanding of the diverse functions of tissue-resident macrophages will provide insights into the mechanisms through which these cells maintain tissue homeostasis and how they may contribute to tissue-specific disease.

Mammary gland development is regulated by numerous systemic and locally derived factors, including hormones, growth factors and cytokines (*Hynes and Watson, 2010*; *Richert et al., 2000*; *Watson and Khaled, 2008*; *Watson et al., 2011*). Tissue-resident macrophages are localized at distinct regions within the mammary gland and have been implicated in various stages of mammary gland development. Early studies demonstrated the presence of resident macrophages in close proximity to epithelial structures, along terminal end buds (TEBs) of developing ductal structures and in the adipose stroma (*Gouon-Evans et al., 2000*; *Schwertfeger et al., 2006*). More recent studies have demonstrated that macrophages are also located directly adjacent to the epithelial cells and can intercalate into the ductal epithelial layer (*Jäppinen et al., 2019*; *Stewart et al., 2019*; *Dawson et al., 2020*). Lineage tracing studies have demonstrated that macrophages in the mammary gland are initially derived from embryonic precursors (*Jäppinen et al., 2019*). Although embryonically derived cells can be found in mammary glands from adult mice, some turnover from blood-derived cells is also observed (*Jäppinen et al., 2019*). Resident macrophages have been functionally linked to mammary gland development. Analysis of mammary glands from mice that are deficient in *Csf-1* demonstrated reduced ductal elongation and branching (*Gouon-Evans et al., 2000*). Furthermore, macrophages that are associated with epithelial ducts have been implicated in regulating stem cell activity (*Gyorki et al., 2009*) and collagen organization (*O'Brien et al., 2010*). In addition, macrophages contribute to mammary epithelial cell turnover during the estrous cycle (*Chua et al., 2010*) and can engulf apoptotic epithelial cells following damage (*Dawson et al., 2020*). Finally, macrophages contribute to epithelial cell death and participate in tissue remodeling during involution following lactation (*Perdiguero et al., 2015*; *Dawson et al., 2020*; *Hughes et al., 2012*).

Macrophages have been implicated in the repair and remodeling of the extracellular matrix (ECM). Macrophages can contribute directly to ECM remodeling through production of proteases and indirectly through modulating fibroblast function (*Kim and Nair, 2019*; *Murray and Wynn, 2011*). In the mammary gland, macrophages promote collagen fibrillogenesis around the developing terminal end bud (*Ingman et al., 2006*). Furthermore, collagen accumulation has been linked to macrophage recruitment during involution (*O'Brien et al., 2010*). Although these studies suggest a role for macrophages in the regulation of ECM within the mammary gland, the specific mechanisms through which macrophages might function in this context have yet to be elucidated. Recent studies have implicated a subpopulation of macrophages that can be identified by Lyve-1 as being involved in the regulation of ECM in the arterial wall and in the lung (*Lim et al., 2018*). Lyve-1 is a receptor for hyaluronan (HA), which is a negatively charged polysaccharide that consists of a repeating disaccharide structure and is an important component of the ECM. HA is important for maintaining tissue structure and hydration and is turned over at high levels under steady state (*Gupta et al., 2019*; *McCourt, 1999*). Hyaluronan synthase (Has2) is expressed by both mammary epithelial cells and stromal cells, and HA has been shown to contribute to epithelial branching in the mammary gland (*Tolg et al., 2017*). However, relatively little is known regarding the localization of HA throughout the adipose stroma and the mechanisms involved in modulating HA homeostasis in the mammary gland.

Here, we describe the localization patterns of a population of cells in the nulliparous mammary gland that is positive for both Lyve-1 and the macrophage marker F4/80. These cells are localized at distinct sites throughout the mammary gland, including in the adipose-associated stroma and in the mammary gland capsule region. Although these cells are also found in association with the epithelial-associated stroma, the majority of F4/80$^+$ cells in this region are negative for the Lyve-1 marker. Further studies demonstrate that the F4/80$^+$Lyve-1$^+$ cells express high levels CD206, which is a marker that is typically associated with tissue reparative macrophages, and that these cells are associated with HA-enriched regions within the mammary gland. Using a shielded bone marrow chimera model, we also demonstrate that these cells are long-lived and capable of self-renewal, which are key traits of tissue-resident macrophages (*Davies et al., 2013*). Analysis of the adipose stroma following macrophage depletion reveals increased levels of HA and collagen, suggesting that macrophages are important for maintaining ECM homeostasis in the mammary stroma. These changes in

the adipose stroma are not accompanied by significant alterations in ductal morphogenesis. These studies identify a distinct stromal macrophage subpopulation within mouse and human mammary glands. Furthermore, our findings suggest that macrophages are important for maintaining ECM homeostasis in mammary glands from virgin mice.

## Results

### Identification of distinct macrophage subpopulations within the mammary gland

Macrophages have been found in distinct locations within the mammary gland, including in close association with epithelial ducts and within the adipose stroma (*Gouon-Evans et al., 2000*; *Schwertfeger et al., 2006*; *Jäppinen et al., 2019*; *Stewart et al., 2019*). The presence of epithelial-associated macrophages is well-documented (*Gouon-Evans et al., 2000*; *Jäppinen et al., 2019*; *Stewart et al., 2019*; *Dawson et al., 2020*), but less is known about the localization of the stromal macrophage populations in the mammary gland. Recently published studies have described markers that can be used to identify specific macrophage populations in other sites. For example, Lyve-1, which is a marker of lymphatic endothelial cells, has been shown to be expressed on tissue-resident macrophages in heart, lung, skin and adipose tissue (*Chakarov et al., 2019*; *Lim et al., 2018*). Lyve-1$^+$F4/80$^+$ cells have been previously characterized in the context of involution, although the localization of these cells in mammary glands from nulliparous mice has not been extensively characterized (*Elder et al., 2018*). Therefore, initial studies were performed to identify the presence of Lyve-1$^+$ macrophages within normal mammary glands from virgin mice. Assessment of Lyve-1$^+$ cells in mouse mammary glands by flow cytometry revealed the presence of distinct populations of CD45$^-$Lyve-1$^+$ cells, which were probably lymphatic endothelial cells, and CD45$^+$Lyve-1$^+$ cells (*Figure 1A*, *Figure 1—figure supplement 1A*) demonstrating the presence of Lyve-1$^+$ leukocytes in the mammary gland. Further analysis was performed to determine the presence of Lyve-1$^+$ macrophages in the mammary gland, which demonstrated that approximately 35% of CD45$^+$CD11b$^+$F4/80$^+$ cells within mammary glands are also Lyve-1$^+$ (*Figure 1B*, *Figure 1—figure supplement 1B*).

To investigate the localization of Lyve-1$^+$ macrophages within the normal mammary gland, mammary gland sections from 6-week-old mice were immunostained for F4/80 and Lyve-1. As expected, Lyve-1 efficiently stained lymphatic endothelial structures associated with the lymph node (*Figure 1—figure supplement 2f*). However, numerous single cells that stained positive for both Lyve-1 and F4/80 were also found to be localized throughout the mammary gland. Therefore, further analysis focused on assessing the association of these cells with terminal end buds (TEBs), the adipose stroma and the mammary capsule. Some F4/80$^+$Lyve-1$^+$ cells were found to be associated with the stroma surrounding the TEBs, but the majority of the F4/80$^+$ cells in this epithelial-associated stroma were found to be Lyve-1$^-$ (*Figure 1C*). We also observed the presence of F4/80$^+$ cells that appear to be intercalated into the epithelial structures, which did not stain positive for Lyve-1 (*Figure 1Ci*). Further analysis of the stromal regions of the mammary gland that were not directly associated with epithelial structures demonstrated that F4/80$^+$Lyve-1$^+$ cells were readily observed within both the adipose stroma and the fibrous capsule surrounding the mammary gland (*Figure 1D,E*, *Figure 1—figure supplement 2a,d*). These findings suggest that F4/80$^+$Lyve1$^+$ cells can be identified in distinct stromal locations throughout the mammary glands of virgin mice.

To confirm the presence of Lyve-1$^+$ macrophages in human mammary glands, we obtained normal human mammary tissue from reduction mammoplasty samples and immunostained for the macrophage marker CD68 and Lyve-1. Analysis of these samples revealed the presence of CD68$^+$Lyve-1$^+$ cells within the interlobular stroma of normal (non-pregnant) human mammary glands (*Figure 1F*). Taken together, these findings demonstrate the presence of Lyve-1$^+$ macrophages within connective-tissue-associated stromal compartments of both mouse and human mammary glands.

### Distinct transcriptional profiles in F4/80$^+$Lyve-1$^+$ and F4/80$^+$Lyve-1$^-$ cells

To determine whether the F4/80$^+$Lyve-1$^+$ and F4/80$^+$Lyve-1$^-$ populations exhibit distinct gene transcriptional profiles, CD45$^+$CD11b$^+$F4/80$^+$Lyve-1$^+$ and CD45$^+$CD11b$^+$F4/80$^+$Lyve-1$^-$ cells were sorted

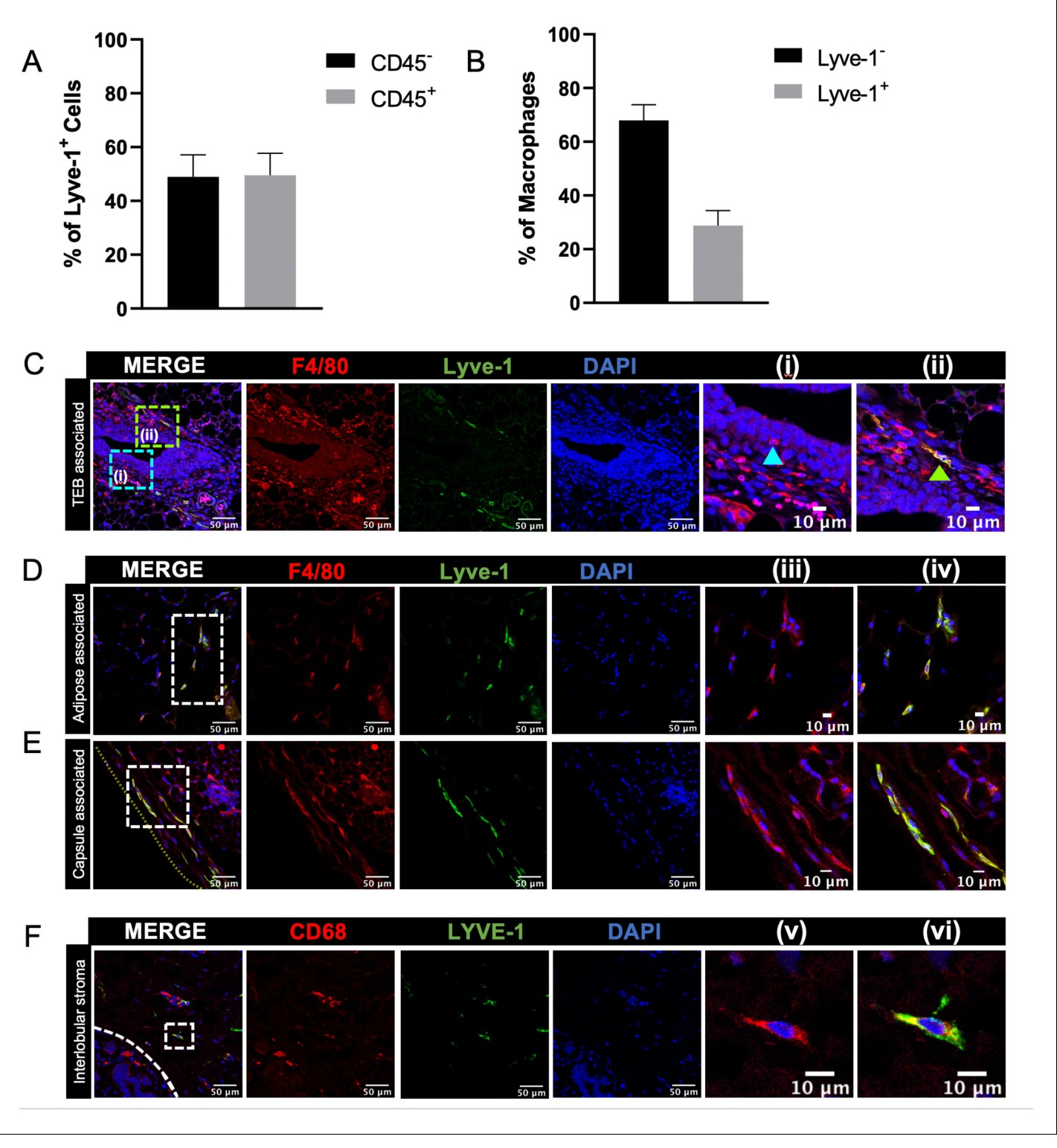

**Figure 1.** Identification of Lyve-1+ macrophages in the mammary gland. (**A**) Mammary glands from 10-week-old mice (n = 4) were assessed for CD45+ and CD45− Lyve-1+ cells by flow cytometry. (**B**) Mammary glands from 10-week-old mice (n = 4) were assessed for CD45+CD11b+F4/80+Lyve-1− and CD45+CD11b+F4/80+Lyve-1+ cells by flow cytometry. (**C**) Mammary glands were harvested from 6-week-old mice, immunostaining was performed for F4/80 and Lyve-1, and the localization of single- and double-positive cells associated with TEBs was examined (n = 5, three images/localization). Representative images of F4/80+Lyve-1− (i, arrowhead) and F4/80+Lyve-1+ (ii, arrowhead) cells are shown. Yellow lines show the margin of the mammary gland. Insets show higher magnification. (**D, E**) Representative images of F4/80+Lyve-1+ cells in the adipose stroma and fibrous capsule of the

*Figure 1 continued on next page*

*Figure 1 continued*

mammary gland. Representative images show F4/80 (iii) and co-staining (iv). Inserts show higher magnification. (**F**) Human mammary glands obtained from reduction mammoplasty samples demonstrate the presence of CD68⁺Lyve-1⁺ macrophages in the interlobular stroma (n = 5, three images/ sample). Representative images show F4/80 (v) and co-staining (vi).

The online version of this article includes the following source data and figure supplement(s) for figure 1:

**Source data 1.** Source data for graphs in panels A and B.
**Figure supplement 1.** Identification of Lyve-1⁺ macrophages in the mammary gland.
**Figure supplement 2.** Localization of Lyve-1⁺ macrophages in the mammary gland.

from mammary glands for bulk RNA-seq analysis (*Figure 2—figure supplement 1A*). Cells were isolated from mammary glands of both 6-week-old and 10-week-old mice. As an initial approach to the analysis, these samples were assessed for gene expression across samples, which allowed for the identification of gene signatures in a way that was independent of stage of ductal development. Gene expression analysis identified 155 genes whose expression was significantly different in these two populations (*Figure 2A* and *Supplementary file 1*). Examination of specific differentially regulated genes demonstrated that the F4/80⁺Lyve-1⁺ cells expressed higher levels of a number of genes that are typically associated with alternatively activated macrophage phenotype and function, including *Mrc1*, *Sparc*, *Gas6*, *Igf1*, *Cd163*, *Apoe*, *Cd209g* and various genes associated with collagen and complement (*Figure 2A*). Notably, many of these genes are typically associated with tissue resolution or reparative functions (*Cui et al., 2020*; *Nepal et al., 2019*; *Novak and Koh, 2013*; *Wynn and Vannella, 2016*). By contrast, genes that were found to be decreased in this population, when compared with the Lyve-1⁻ population, included genes that are associated with inflammation and antigen presentation including *Il1b*, *Cd74* and *H2-Ab1* (*Figure 2A* and *Supplementary file 1*).

To verify the presence of Lyve-1⁺ macrophages in the normal mammary gland independently, single-cell RNA-seq (scRNA-seq) analysis was performed on immune cells isolated from mammary glands of 10-week-old virgin mice. For these studies, CD45⁺ cells were isolated from mammary glands of diestrous-staged 10-week-old mice for further analysis by scRNA-seq. Differential gene expression analysis identified 10 distinct subpopulations (*Figure 2B,C*, *Supplementary file 2*). Clusters 0 and 4 are enriched in macrophage-related markers, including *Cd68*, *Csf1r*, *Cd14* and *Mafb* (*Figure 2D* and *Figure 2—figure supplement 1B*). Clusters 1 and 2 are enriched in genes associated with T cells (*Cd3d*, *Cd8b1*) and B cells (*Igkc*, *Ms4a1*). Clusters 3 and 7 express genes associated with dendritic cells including *Ccl22* and *Xcr1*, respectively. Cells associated with cluster 5 express NK-cell-related genes (*Ncr1*, *Nkg7*). Cluster 6 represents a proliferative population with the expression of genes such as *Mki67* and *Stmn1*. Cells in cluster 8 are associated with the expression of the monocyte marker *Ly6c2*. Finally, Cluster 9 appears to represent a small population of contaminating epithelial cells (*Prlr*, *Krt18*). Further examination of genes that are associated with clusters 0 and 4 demonstrated enrichment in *Lyve1* expression in cluster 4, as well as enrichment of the expression of additional genes identified in the RNA-seq data including *Cd209g* and *Gas6* (*Figure 2E*). Recently published studies identified three distinct macrophage subpopulations, which include CD11c⁺ ductal macrophages and two CD11cˡᵒCD11b⁺ stromal populations (*Dawson et al., 2020*). CD11b (*Itgam*) was enriched within clusters 0 and 4 (*Figure 2—figure supplement 1C*), suggesting that these populations may correspond with the two CD11b⁺ stromal populations. Notably, one of the stromal subpopulations was found to be Lyve-1ʰⁱ (*Dawson et al., 2020*). CD11c was not detectable at appreciable levels in any subpopulation, possibly as the result of dropout in our analysis (*Figure 2—figure supplement 1C*). However, we confirmed that Lyve-1 expression is low on CD11bˡᵒ macrophages (*Figure 2—figure supplement 1D*), consistent with the findings that Lyve-1 is expressed primarily on CD11b⁺ macrophages in the mammary gland.

To assess the similarities between the Lyve-1⁺ population characterized by bulk RNA-seq analysis and the cluster 4 population of macrophages, we performed gene set enrichment analysis (GSEA). Differentially up- and downregulated gene sets from the bulk RNA-seq Lyve-1⁻ and Lyve-1⁺ populations were used for GSEA of the single cell cluster 4 vs 0 differentially expressed rank list. This analysis demonstrated strong, directionally concordant enrichment of gene expression between the bulk RNA-seq Lyve-1⁺ population and the scRNA-seq cluster 4 population (*Figure 2F*,

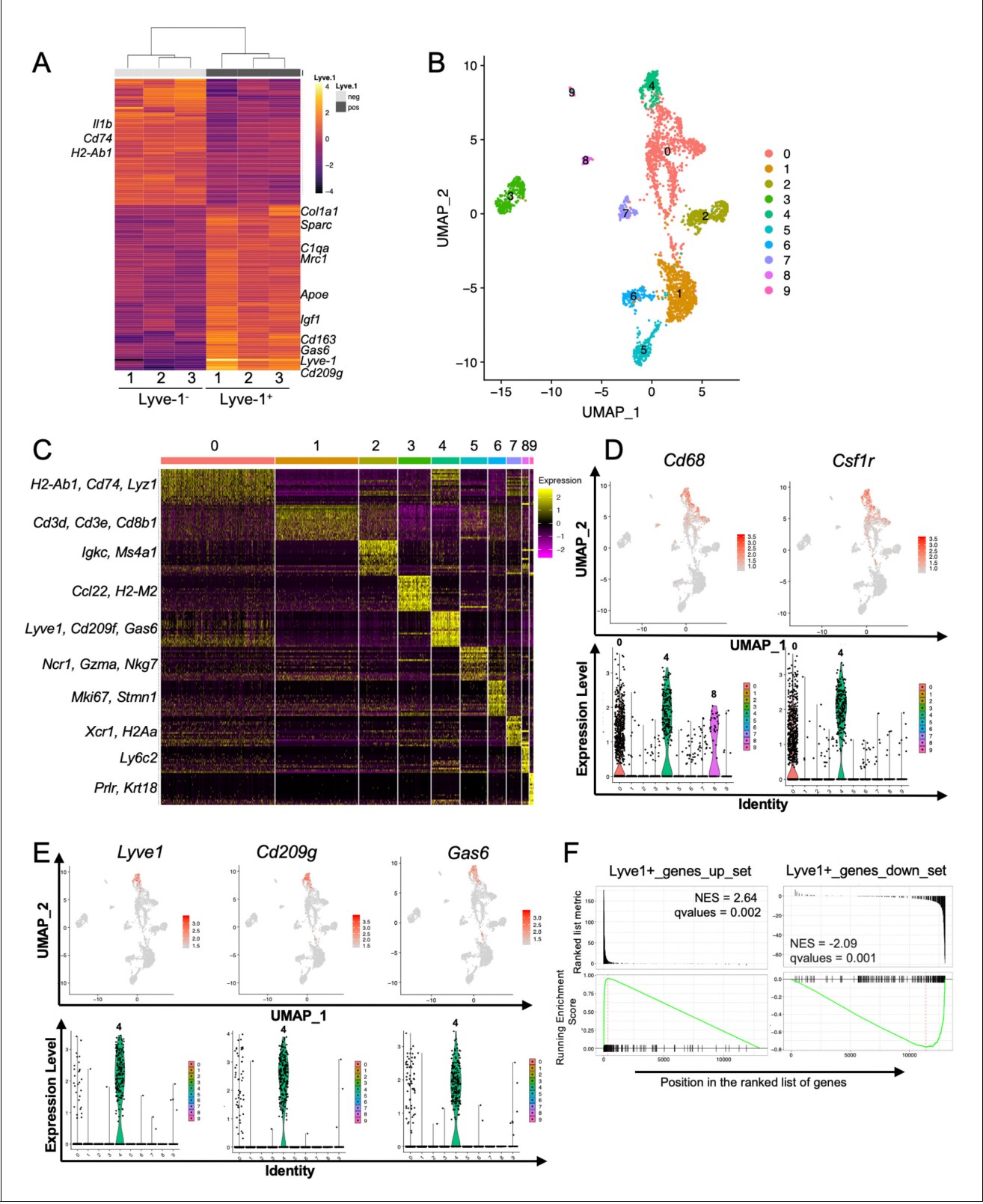

**Figure 2.** Identification of a distinct Lyve-1+ macrophage subpopulation by transcriptional profiling. (**A**) Heat map of RNA-seq analysis of CD45+CD11b+F4/80+Lyve-1− and CD45+CD11b+F4/80+Lyve-1+ cells isolated from 6-week-old (lane 1) and 10-week-old (lanes 2–3) mice. Genes shown have an adjusted p-value <0.01 and a fold-change of >1 and <10 (abs values). For each lane, n = 4 pooled mice. (**B**) UMAP of scRNA-seq analysis of CD45+ cells isolated from 10-week-old mice as generated by Seurat. (**C**) Heat map of the top 20 differentially regulated genes in each cluster. (**D**)

*Figure 2 continued on next page*

Figure 2 continued

Feature plots and violin plots of selected macrophage genes. (E) Feature plots and violin plots of genes associated with cluster four that were also found in the bulk RNA-seq analysis. (F) GSEA demonstrating that the single-cell populations (clusters) are enriched for in the data from the bulk RNA-seq analysis.

The online version of this article includes the following figure supplement(s) for figure 2:

**Figure supplement 1.** Identification of a distinct Lyve-1$^+$ macrophage subpopulation by transcriptional profiling.

Supplementary file 3). These results demonstrate similarities between the Lyve-1$^+$ populations in both the bulk RNA-seq and the single-cell RNA-seq analysis. Therefore, two independent transcriptional-profiling approaches demonstrate the presence of a distinct Lyve-1$^+$ population of macrophages within the normal mammary gland.

## F4/80$^+$Lyve-1$^+$ cells are enriched for CD206 expression

*Mrc1*, which encodes for CD206, was found to be enriched in the Lyve-1$^+$ macrophage population in the bulk RNA-seq analysis (*Figure 2A*) and in the scRNA-seq analysis (*Figure 3A*). CD206 expression is often found on tissue-resident macrophage populations and is typically associated with a tissue-reparative phenotype (*Novak and Koh, 2013*). Therefore, further studies were performed to determine the colocalization of Lyve-1 with CD206 in the mammary gland. Initial analysis of mammary glands using flow cytometry demonstrated that approximately 90% of F4/80$^+$Lyve-1$^+$ cells are also positive for CD206 (*Figure 3B*, *Figure 3—figure supplement 1*). We also found that approximately 45% of F4/80$^+$CD206$^+$ cells are positive for Lyve-1 (*Figure 3B*). These results demonstrate that populations of both CD206$^+$Lyve-1$^+$ and CD206$^+$Lyve-1$^-$ cells are present in the mammary gland, which is consistent with the scRNA-seq data (*Figure 3A*). To further confirm these findings, immunofluorescence was performed to localize F4/80$^+$CD206$^+$ cells in the mammary gland. As with F4/80$^+$Lyve-1$^+$ staining, we observed F4/80$^+$CD206$^+$ cells within the epithelial-associated stroma surrounding the TEB, the adipose stroma and the mammary capsule (*Figure 3C–E*). Although not all F4/80$^+$ cells associated with the epithelial-associated stroma were CD206$^+$, more CD206$^+$ cells were present within this region than were observed with the Lyve-1$^+$ staining (*Figure 3C*). However, as for the Lyve-1 staining, the F4/80$^+$ cells intercalated within the epithelium were found to be negative for CD206 (*Figure 3C*). These results confirm that Lyve-1$^+$ cells are predominantly positive for CD206, although a distinct population of CD206-expressing macrophages is also present in the mammary gland. Previously published studies have demonstrated the presence of two distinct macrophage populations in the mammary gland with variable intensity of expression of F4/80 and CD206; these F4/80$^{hi}$,CD206$^{hi}$ and F4/80$^{int}$,CD206$^{neg/low}$ populations were found to correspond with fetal-derived and bone marrow-derived macrophages, respectively (*Jäppinen et al., 2019*). On the basis of these studies, CD206 was suggested to represent a marker of resident macrophages in the mammary gland. Consistent with these previously published findings, we also found that the majority of the F4/80$^{hi}$ cells express CD206 (*Figure 3F*) and that the majority of F4/80$^+$Lyve-1$^+$ population cells are associated with the F4/80$^{hi}$ population (*Figure 3G*). These findings suggest that Lyve-1$^+$ macrophages represent a stromal-resident macrophage subpopulation in the mammary gland that is associated with a CD206$^+$F4/80$^{hi}$ phenotype.

Lyve-1$^+$ macrophages in the mammary gland exhibit low rates of turnover under steady state and are capable of self-renewal. Tissue-resident macrophages in different organ sites show varying levels of turnover and self-renewal (*Ginhoux and Guilliams, 2016*). Studies were performed to characterize the turnover and self-renewal properties of Lyve-1$^+$ macrophages in adult mammary glands using a shielded bone marrow chimera approach (*Misharin et al., 2017*). BALB/c CD45.2 mice underwent lethal irradiation (800cGy, split doses) with the mammary glands shielded using lead to maintain the presence of resident tissue macrophages in the mammary gland, followed by treatment with busulfan (25 mg/kg) to deplete any residual bone-marrow-derived cells. Subsequently, irradiated mice were transplanted with bone marrow from BALB/c CD45.1 mice (*Figure 4A*). After 6 weeks, chimerism in the blood was examined and the proportion of CD45.1 and CD45.2 macrophages in the mammary glands was assessed and defined as donor and host, respectively. Unlike the previous experiments, this experiment did not allow us to differentiate between F4/80$^{hi}$ and F4/80$^{int}$ macrophages in the bone marrow chimera studies. This could be due to differences in experimental

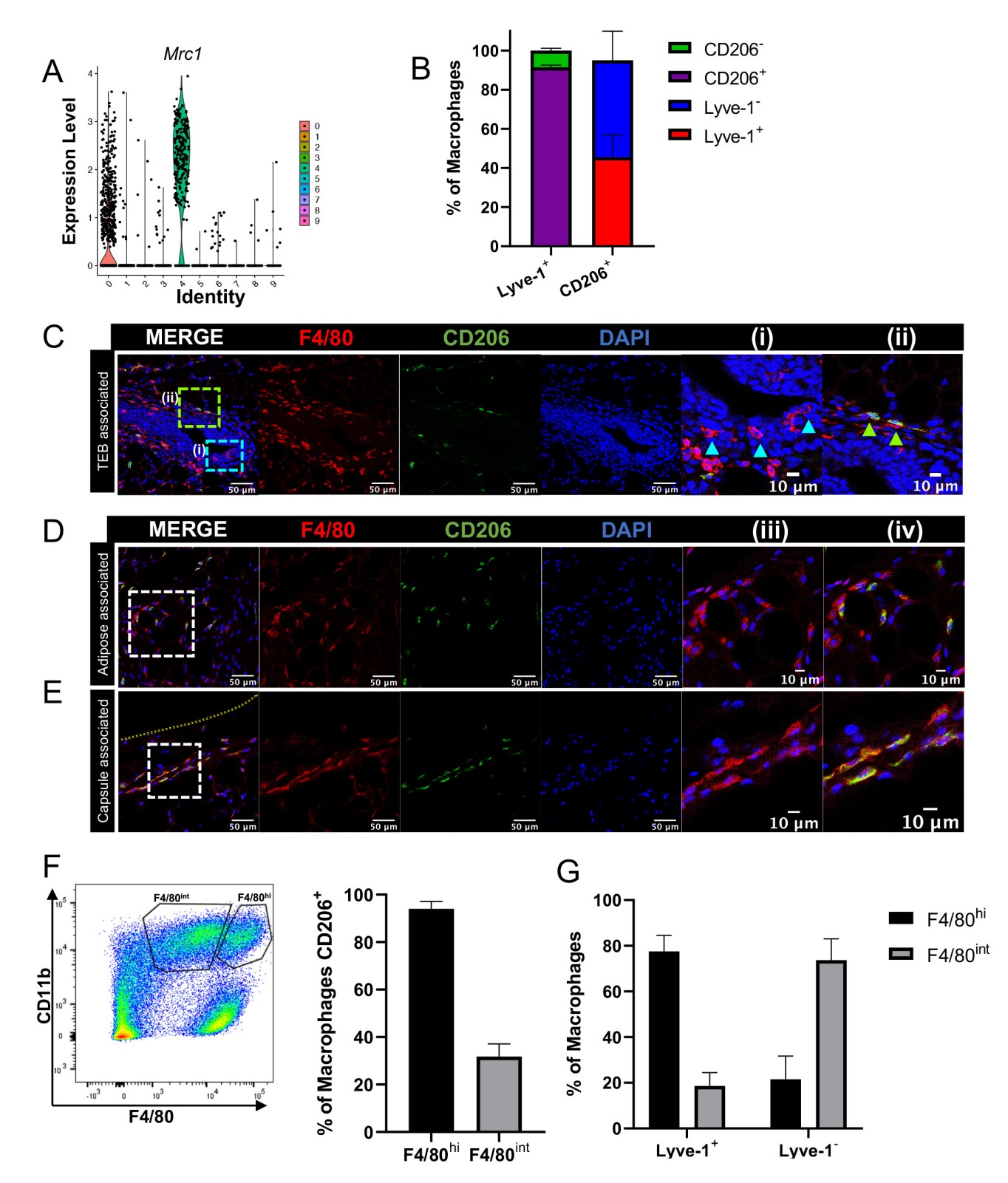

**Figure 3.** F4/80+Lyve-1+ cells are enriched for CD206 expression. (A) Violin plot of *Mrc1* expression in the single-cell RNA-seq dataset shown in *Figure 2*. (B) Mammary glands from 10-week-old mice (n = 4) were assessed for CD206 expression in CD45+CD11b+F4/80+Lyve-1- and CD45+CD11b+F4/80+Lyve-1+ cells by flow cytometry. (C) Mammary glands were harvested from 6-week-old mice (n = 5, three images/localization), immunostaining was performed for F4/80 and CD206, and the localization of single- and double-positive cells associated with TEBs was

*Figure 3 continued on next page*

*Figure 3 continued*

examined. Representative images of F4/80$^+$CD206$^-$ (i, arrowheads) and F4/80$^+$CD206$^+$ (ii, arrowheads) cells are shown. (D, E) Representative images of F4/80$^+$CD206$^+$ cells in the adipose stroma and fibrous capsule of the mammary gland. Representative images show F4/80 (iii) and co-staining (iv). The yellow line in panel (E) shows the margin of the mammary gland. Insets show higher magnification. (F) Flow cytometry analysis showing F4/80$^{hi}$ and F4/80$^{int}$ cells in the mammary gland and CD206 expression in each population. (G) Flow cytometry analysis of F4/80 expression on Lyve-1$^-$ and Lyve-1$^+$ cells. n = 4 mice analyzed.

The online version of this article includes the following source data and figure supplement(s) for figure 3:

**Source data 1.** Source data for graphs in panels B, F, and G.
**Figure supplement 1.** F4/80$^+$Lyve-1$^+$ cells are enriched for CD206 expression.

protocol, such as mouse strain (BALB/c mice were used rather than FVB/n mice because of the availability of congenic strains), or to the systemic effects of irradiation. Although 95% donor chimerism was established in circulating monocytes in the blood, only ~30% of the macrophages in the mammary gland were of donor origin, suggesting that the majority of resident macrophages were maintained in the mammary gland over the 6-week time course (*Figure 4B*). Furthermore, the majority of Lyve-1$^+$ macrophages were host-derived, demonstrating a low level of replenishment of F4/80$^+$Lyve-1$^+$ cells from bone-marrow-derived sources in the mammary gland at steady state (*Figure 4C*). This finding is consistent with recently published studies that also demonstrate a low rate of turnover of stromal macrophages in the mammary gland (*Dawson et al., 2020*). To further assess the ability of these cells to self-renew, mice were pulsed with BrdU for 2 hr and immunofluorescence was performed to identify BrdU$^+$Lyve-1$^+$ cells in the mammary gland. BrdU$^+$Lyve-1$^+$ double-positive cells were identified close to the epithelial-associated stroma and in the adipose stroma (*Figure 4D*). Taken together, these studies suggest that Lyve-1$^+$ macrophages show low rates of turnover in the mammary gland at steady state and that a proportion of these cells are capable of self-renewal.

## Lyve-1$^+$ cells are associated with hyaluronan-enriched regions in the mammary gland and in mammary tumors

Lyve-1 is a well-established receptor for hyaluronan (HA), a key component of the extracellular matrix (*Banerji et al., 1999*). HA is present in the normal mammary gland and is known to be important for epithelial branching (*Tolg et al., 2017*). However, whether macrophages are specifically associated with HA-containing regions in the mammary gland has not been examined. Initial studies were performed to examine HA localization in the mammary gland by immunostaining sections with HA binding protein (HABP), along with smooth muscle actin (SMA) as a marker of myoepithelial cells to visualize the ducts. HA was found to be present within the epithelial-associated stroma along TEBs (*Figure 5A*). In addition, HA was found to be associated with fibrous septae tracking through the adipose tissue and was also present at high levels in the fibrous capsule surrounding the mammary gland (*Figure 5A,B*). To determine whether Lyve-1$^+$ cells are significantly associated with HA-rich regions of the mammary gland, mammary glands were co-stained with HABP and Lyve-1. Despite the presence of HA surrounding the TEB, few Lyve-1$^+$F4/80$^+$ cells are found within the TEB-associated stroma (*Figure 1C*), therefore, we focused on the ECM-rich regions within the adipose stroma. Lyve-1$^+$ cells were found to be primarily associated with HA-containing structures in both the adipose stroma and in the mammary gland capsule region (*Figure 5B*). Quantification of Lyve-1$^+$ cells demonstrated that they preferentially bind to HA-rich regions in both the adipose stroma and the fibrous capsule (*Figure 5C*). To further confirm that these cells represent the Lyve-1$^+$ macrophage population, mammary gland sections were co-stained with F4/80, Lyve-1 and HABP. Analysis of co-staining demonstrated that the HA-associated Lyve-1$^+$ cells are positive for F4/80 (*Figure 5—figure supplement 1*).

To assess the localization of Lyve-1$^+$ cells in human mammary gland, tissue sections were stained with HABP and Lyve-1. HA was abundantly present in the human mammary gland (*Figure 5D*). As in the mouse mammary gland, Lyve-1$^+$ cells were found to be associated with fibrous septae within the stroma (*Figure 5D*). Together, these studies demonstrate the presence of HA within various areas of connective tissue throughout the mammary gland and demonstrate that Lyve-1$^+$ cells specifically associate with these HA-enriched regions, which is consistent with the function of Lyve-1 as a receptor for HA. However, further studies are required using models in which Lyve-1 is selectively ablated

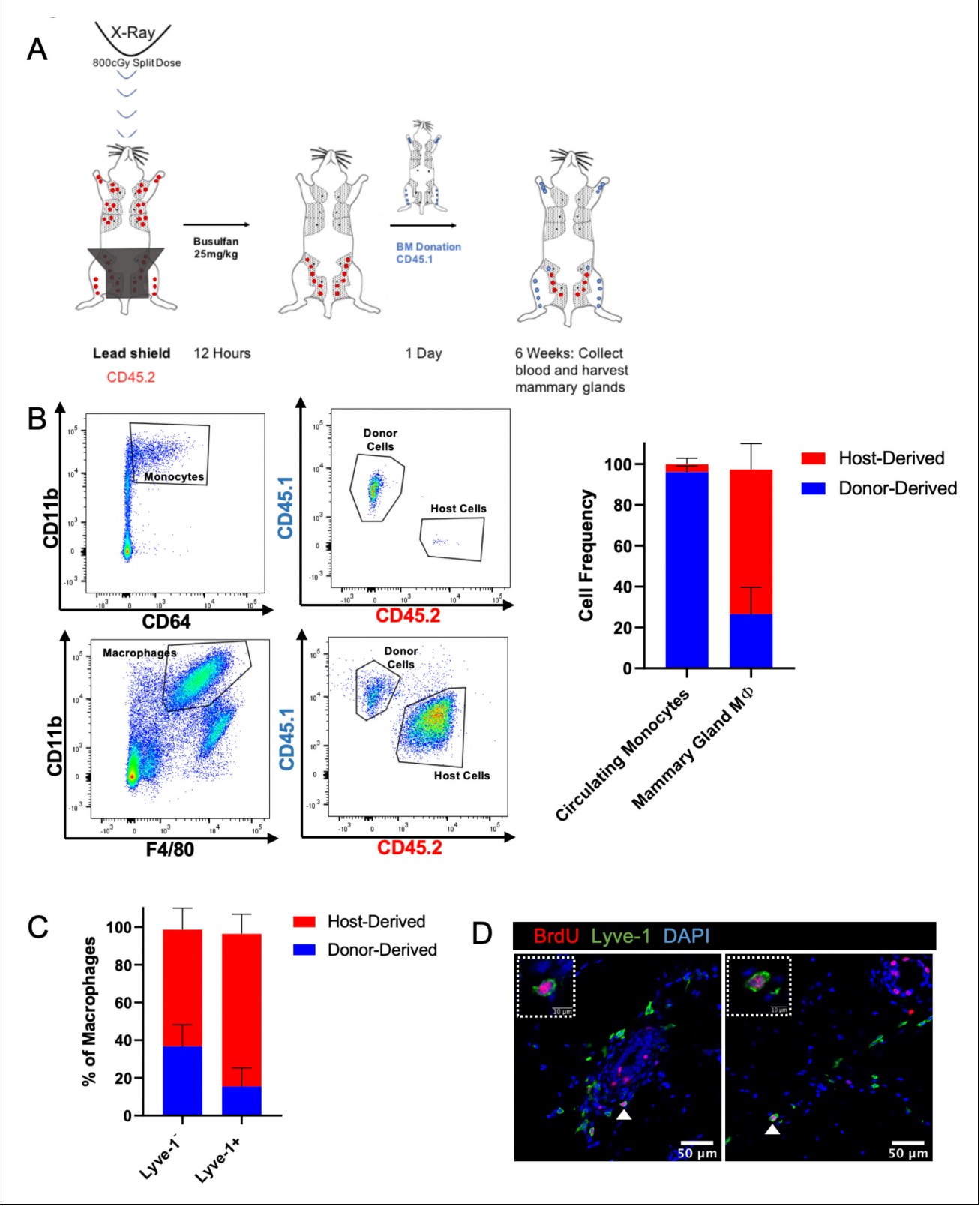

**Figure 4.** Lyve-1[+] macrophages exhibit low turnover and self-renewal in the mammary gland. (**A**) Schematic of shielded bone marrow chimera experiment. (**B**) Representative flow cytometry plots showing the gating and identification of host (CD45.2) and donor (CD45.1) cells in the blood and mammary gland. Quantification of flow cytometry demonstrates that although most of the monocytes in the blood are donor-derived, the majority of macrophages in the mammary gland are host-derived. Cells were first gated on Live and pan-CD45 expression. In the quantification graphs, n = 23

*Figure 4 continued on next page*

*Figure 4 continued*

mice from three separate experiments. (C) Quantification of flow cytometry demonstrates that the majority of Lyve-1$^+$ macrophages remain host-derived 6 weeks after bone marrow transplant. (D) Mammary glands were isolated from mice after a 2 hr BrdU pulse and immunostained for BrdU and Lyve-1 (n = 3, three images/localization). Examples of BrdU$^+$Lyve-1$^+$ cells are shown, insets show higher magnification.

The online version of this article includes the following source data for figure 4:

**Source data 1.** Source data for graphs in panels B and C.

in macrophages to demonstrate the importance of this receptor in the localization of these cells to the HA-containing ECM.

Given the well-established importance of macrophages in the tumor microenvironment, we assessed the localization of Lyve-1$^+$ macrophages in mammary tumors. 4T1 cells represent a well-characterized mammary tumor model of triple negative breast cancer that are injected into the fat pads of syngeneic BALB/c mice (*Aslakson and Miller, 1992*). Although F4/80$^+$Lyve-1$^+$ cells were found at low levels within the tumor parenchyma, these cells were found to be enriched within the peri-tumoral stroma (*Figure 5E*). Staining for both Lyve-1 and HA demonstrated that these Lyve-1$^+$ cells are embedded within HA in the peri-tumoral stroma (*Figure 5F*). These findings demonstrate that Lyve-1$^+$ macrophages represent a distinct subset of macrophages that localize to specific stromal regions within both the normal mammary gland and the tumor microenvironment.

## Macrophage depletion impacts ECM in the stroma

Given the localization of the Lyve-1$^+$ cells to fibrous stroma in the adipose tissue, functional studies were performed to determine whether macrophages are important for maintaining these structures under steady state. To deplete macrophages, mice were treated with pexidartinib (45 mg/kg), which is an inhibitor of CSF1R, c-Kit and Flt3 that is commonly used to deplete resident macrophages (*DeNardo et al., 2011*), by daily oral gavage. Treatment was initiated at 5 weeks of age and macrophage depletion was maintained for 2 weeks. Mammary glands were removed and assessed by flow cytometry to confirm macrophage depletion. Analysis of the bulk macrophage population revealed a significant reduction in the total macrophage cell counts (*Figure 6A*). Further assessment of Lyve-1$^+$ and Lyve-1$^-$ populations demonstrated that both of these populations were significantly reduced following CSF1R inhibition (*Figure 6A*). Further studies were performed to assess the effects of macrophage depletion on ductal elongation and ECM modulation in the stromal compartment. Analysis of epithelial structures revealed no significant alterations in ductal elongation or in branching following macrophage depletion (*Figure 6—figure supplement 1*). However, analysis of HA in the stromal regions of the mammary glands revealed increased levels of HA within the adipose stroma following macrophage depletion (*Figure 6B,C*). These findings were further confirmed by assessing HA levels in mammary gland lysates by ELISA (*Figure 6D*). Because macrophages have also been linked to maintaining collagen turnover (*Lim et al., 2018*; *Madsen et al., 2013*), levels of collagen in the mammary stroma were also assessed. Furthermore, assessment of collagen by trichrome staining demonstrated increased collagen within the adipose stroma (*Figure 6B,C*), suggesting that loss of macrophages leads to alterations in the ECM-enriched regions in the adipose stroma of the mammary gland. Further confirming the hypothesis that Lyve-1$^+$ macrophages regulate ECM, GSEA of the scRNA-seq data demonstrated a significant enrichment in gene sets associated with ECM, endocytosis, and glycosaminoglycan binding in cluster 4 compared with cluster 0 (*Figure 6E,F*). Interestingly, upregulation of specific genes within the glycosaminoglycan binding signature included genes involved in HA binding, internalization and degradation, such as Lyve1 and Hyal2 (*Supplementary file 4*). Taken together, these results suggest that a key function of resident macrophages under steady state conditions is to modulate ECM turnover in the mammary gland stroma.

## Discussion

Here, we demonstrate the presence of a subpopulation of resident macrophages in the normal mammary gland that is specifically associated with ECM-rich connective tissue located within the

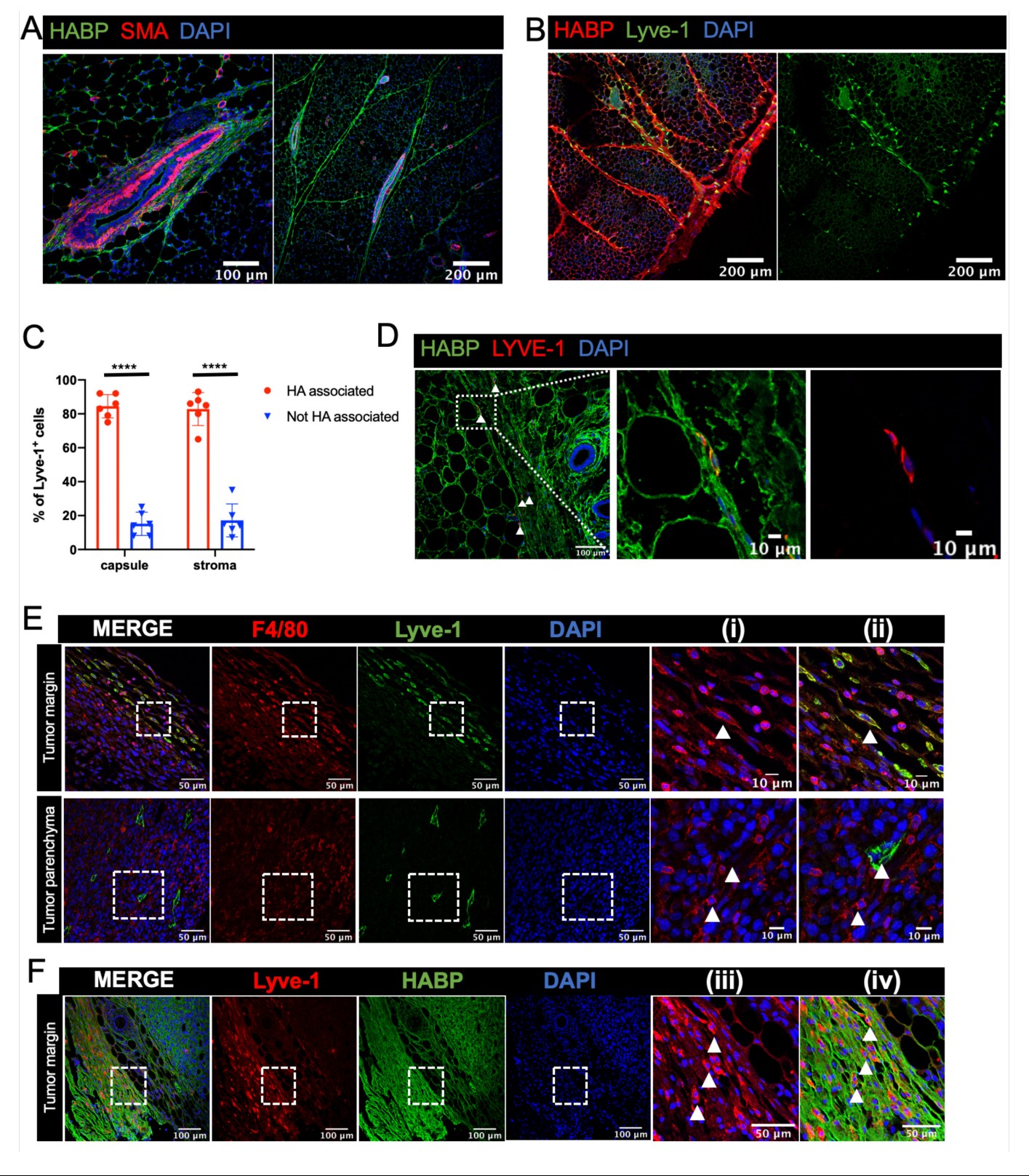

**Figure 5.** Lyve-1[+] cells localize to HA-enriched ECM in the mammary gland. (**A**) Mammary glands from 6-week-old mice were immunostained for HA using HA binding protein (HABP, green) and smooth muscle actin (SMA, red) (n = 3, three images/sample). (**B**) Mammary glands from 6-week-old mice were immunostained for HA using HA binding protein (HABP, red) and Lyve-1 (green) showing staining of the fibrous capsule and fibrous septae in the

*Figure 5 continued on next page*

*Figure 5 continued*

adipose stroma. (**C**) Quantification of Lyve-1$^+$ cells associated with the capsule and adipose stromal regions in the mammary gland. N = 6 mice, three images analyzed per mammary gland. ****p<0.0001. (**D**) Co-staining of HABP and Lyve-1 in human mammary gland demonstrating the presence of HA-associated Lyve-1$^+$ cells (n = 3, five images/sample). Arrowheads show Lyve-1$^+$ cells and inserts show higher magnification. (**E**) Mammary tumor sections from 4T1 tumors were immunostained for F4/80 (red) and Lyve-1 (green) (n = 4, three images/localization). Representative images are shown of the peri-tumoral stroma and the tumor parenchyma. Inserts show higher magnification. (**F**) Mammary tumor sections from 4T1 tumors were immunostained for Lyve-1 (red) and HABP (green). Insets show higher magnification.

The online version of this article includes the following source data and figure supplement(s) for figure 5:

**Source data 1.** Source data for graph in panel C.
**Figure supplement 1.** Lyve-1$^+$ cells localize to HA-enriched ECM in the mammary gland.

stroma of both mouse and human mammary glands. Previous studies of resident macrophages in the mammary gland have focused primarily on macrophages that are located in close proximity to epithelial structures, and that demonstrated their presence adjacent to TEBs during ductal elongation and in close association with developing alveolar buds during pregnancy (*Gouon-Evans et al., 2000*; *Schwertfeger et al., 2006*). The use of mice that are deficient for *Csf1*, which lack mature macrophages, highlighted the importance of these cells in the context of mammary gland development (*Gouon-Evans et al., 2000*). More recent studies have also identified distinct myeloid cell populations, including CD11c$^+$ antigen presenting cells, and Csf1r$^+$ macrophages that are localized in close association with ductal epithelial cells (*Stewart et al., 2019*; *Plaks et al., 2015*). Macrophages that are localized within the mammary stroma have been reported, but recent studies have highlighted that these macrophages represent a subpopulation that is distinct from epithelial-associated macrophages (*Gouon-Evans et al., 2000*; *Schwertfeger et al., 2006*; *Dawson et al., 2020*). We demonstrate here that a distinct population of Lyve-1$^+$ macrophages is associated with ECM-rich regions within the mammary stroma and the fibrous capsule of the mammary gland. Furthermore, we demonstrate that resident macrophages are important for maintaining homeostasis of HA, a key component of the ECM, within these regions. Further studies are warranted using models in which Lyve-1$^+$ macrophages can be selectively depleted to define the importance of this subpopulation in ECM homeostasis within the adipose stroma.

Investigations into the ontogeny, longevity and function of distinct tissue-resident macrophage populations have demonstrated that resident macrophages can be derived from embryonic populations and self-sustain within the tissue. These studies have also found that turnover of resident macrophages in adults can vary depending upon the tissue site (*Ginhoux and Guilliams, 2016*). Recent studies have demonstrated that macrophages in the mammary gland are initially seeded from embryonic precursors and that these macrophages are maintained into adulthood, with embryonically derived macrophages remaining detectable at 3 months of age (*Jäppinen et al., 2019*). Furthermore, the authors demonstrated that the embryonically derived population was associated with an F4/80$^{hi}$ and CD206$^{hi}$ profile. Interestingly, we found that the Lyve-1$^+$ population also exhibits a F4/80$^{hi}$ and CD206$^+$ profile, although not all of the F4/80$^{hi}$ and CD206$^+$ populations were found to be Lyve-1$^+$, providing further support for distinct resident macrophage populations in the mammary gland. Furthermore, analysis of BrdU incorporation revealed that the Lyve-1$^+$ cells are capable of self-renewal. Studies of resident macrophages in the lung have demonstrated that perivascular Lyve-1$^+$ macrophages are derived from Ly6C$^{hi}$ monocytes (*Chakarov et al., 2019*). The majority of Lyve-1$^+$ macrophages were maintained during the 6-week bone marrow chimera experiment, but a small population of donor-derived Lyve-1$^+$ macrophages was also detected, suggesting a slow rate of turnover of this population from the blood. Therefore, it is possible that this population represents a combination of infiltrating and self-renewing macrophages.

Through co-localization studies, we found that the majority of Lyve-1$^+$ cells were associated with HA-containing regions within the mammary stroma, including the fibrous capsule surrounding the mammary gland. The mammary gland capsule is comprised of a fascial plane that is likely to serve a structural function to encase and protect the fat pad. We do not yet know whether this structure also serves a barrier function that reduces pathogen infiltration, as has been shown to be the case

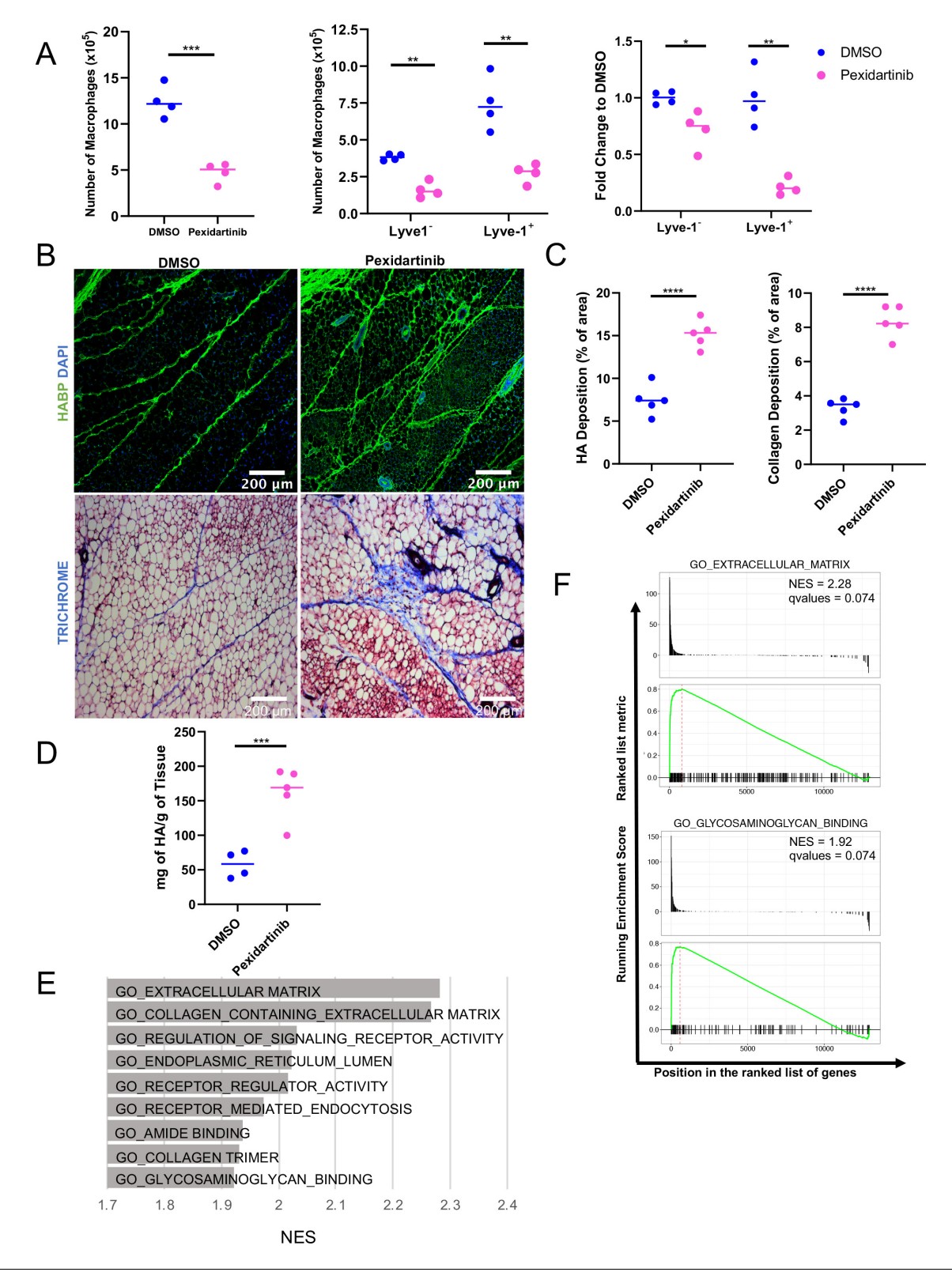

**Figure 6.** Macrophage depletion impacts ECM in the stroma. (**A**) 5-week-old female mice were treated with pexidartinib for 2 weeks and mammary glands were harvested for analysis of macrophages for CD45⁺CD11b⁺F4/80⁺ by flow cytometry. Flow cytometry demonstrated depletion of CD45⁺CD11b⁺F4/80⁺Lyve-1⁻ and CD45⁺CD11b⁺F4/80⁺Lyve-1⁺ cells in the mammary gland following pexidartinib treatment. Quantification of total cell counts and fold-change depletion are shown. Total Macrophages Number DMSO vs pexidartinib, p=0.0007; Lyve1⁻ Macrophages Number DMSO vs

*Figure 6 continued on next page*

*Figure 6 continued*

pexidartinib, p=0.0015; Lyve1$^+$ Macrophages Number DMSO vs pexidartinib, p=0.0095; Lyve1$^-$ Fold Change DMSO vs pexidartinib, p=0.037; Lyve1$^+$ Fold Change DMSO vs pexidartinib, p=0.0052. These data demonstrate one representative experiment, experiments were repeated three times with similar results, n = 4 mice per treatment group. (B) Mammary glands from mice treated with either pexidartinib or solvent control were stained for HA (HABP, green) or with trichrome to visualize collagen-containing ECM (collagen, blue). (C) Quantification of HABP and collagen staining. Between 3 and 6 images/sample were analyzed for HABP and collagen staining, n = 5 per treatment group. Three replicates were used for HABP analysis and one replicate was done for collagen analysis. HA deposition (% of area) DMSO vs pexidartinib, p<0.0001 (pooled) and collagen deposition (% of area) DMSO vs pexidartinib, p<0.0001. (D) Quantification of HA in mammary glands isolated from mice treated with pexidartinib or solvent control normalized to gland weight, p=0.0015, n = 4 mice for control group and n = 5 mice for pexidartinib group. (E) GSEA analysis of the C5 database for gene sets enriched in cluster 4 from the scRNA-seq analysis showing the top nine enriched gene sets by NES. (F) GSEA plots of representative lists identified in panel (E). *, p<0.05; **, p<0.01; ***, p<0.005; ****, p<0.0001; using Welch's t-test.

The online version of this article includes the following source data and figure supplement(s) for figure 6:

**Source data 1.** Source data for graphs in panels A, C, and D.
**Figure supplement 1.** Macrophage depletion does not impact ductal elongation of branching.

for macrophage-enriched capsular structures surrounding intraperitoneal organs such as the liver (*Sierro et al., 2017*). Interestingly, while some F4/80$^+$Lyve-1$^+$ cells were found to be associated with the stroma in direct association with TEBs, the majority of F4/80$^+$ cells in this region were negative for Lyve-1 staining. Results from the RNA-seq data suggest that the F4/80$^+$Lyve-1$^-$ population is enriched in pro-inflammatory factors when compared with the F4/80$^+$Lyve-1$^+$ population, suggesting a potential role in immunosurveillance. This finding is consistent with the localization of the F4/80$^+$Lyve-1$^-$ cells near the epithelial structures, which have the potential for pathogen exposure via the ductal lumen. It is important to note that the bulk RNA-seq analysis was performed on CD11b$^+$ cells, thus this analysis probably does not include the recently described CD11c$^+$ ductal macrophages (*Dawson et al., 2020*). It would be interesting to determine where the CD11b$^+$F4/80$^+$Lyve-1$^-$ cells are localized in comparison with the CD11c$^+$ ductal macrophages. In addition, although the majority of the F4/80$^+$Lyve-1$^+$ cells were positive for CD206, we also observed a separate population of CD206$^+$Lyve-1$^-$ cells, suggesting that CD206 expression is not limited to the Lyve-1$^+$ macrophage population. Overall, these findings demonstrate the presence of distinct subpopulations of macrophages in the mammary gland that probably exhibit distinct functions that are based on their localization and exposure to tissue-specific signals.

Macrophages have been implicated in ECM repair and remodeling during the resolution phase of wound healing and in the tumor microenvironment (*Condeelis and Pollard, 2006*; *Wynn and Barron, 2010*). In the mammary gland, macrophages have been shown to modulate the organization of type I collagen around the TEB (*Ingman et al., 2006*). ECM regulation is complex and macrophages can impact levels of ECM within a tissue via a variety of mechanisms, including direct production of ECM molecules, expression of ECM-modulating proteases and regulation of ECM-producing fibroblasts. Gene-profiling studies have demonstrated that macrophages express genes that are associated with ECM remodeling, including ECM molecules and proteases (*Etich et al., 2019*). Macrophages have also been shown to produce proteases that can act directly on ECM factors, such as collagen, to maintain ECM homeostasis and suppress fibrosis (*Lim et al., 2018*). Finally, macrophages can impact fibroblast-mediated production of ECM, which contributes to the modulation of fibrosis (*Wynn and Barron, 2010*). Lyve-1$^+$ macrophages have been suggested to repress fibrosis directly, in part through production of collagen cleaving proteases (*Lim et al., 2018*). Given that Lyve-1 is a well-documented receptor for HA (*Banerji et al., 1999*; *Schwertfeger et al., 2015*), we hypothesized that Lyve-1 expressing macrophages localize to HA-containing regions in the mammary gland. HA was found to be enriched around TEBs, in fibrous septae within the adipose stroma and in the capsule surrounding the fat pad. Despite the localization of HA within the TEB stroma, few Lyve-1$^+$ cells were found within this region. However, Lyve-1$^+$ cells were found to be associated with other HA-enriched regions, including the fibrous septae and the capsule. While the reasons for this are unknown, differences in the composition of the TEB-associated stroma and the adipose-associated stroma may drive expression of unique ECM receptors and/or recruit distinct

macrophage subtypes that have different ECM binding affinities. Future studies are required to determine whether Lyve-1 expression is required for the localization of these macrophages to HA-enriched regions within the mammary stroma.

HA is a major component of connective tissue, and high molecular weight HA is important for maintaining tissue structure and hydration. HA turnover is estimated at approximately 15 g per day, suggesting high levels of turnover in tissues under steady state (*Stern, 2004*). The specific mechanisms involved in the regulation of HA turnover in normal tissues remain unknown, but macrophages have been implicated in HA catabolism on the basis of their ability to internalize HA via endocytosis (*Underhill et al., 1993*). We also found that macrophage depletion leads to increased collagen accumulation. Although macrophages have been linked to collagen degradation via production of MMPs, *Mrc1*-expressing macrophages have also been shown to internalize and degrade collagen in the dermis (*Madsen et al., 2013*). Further studies involving the quantification and analysis of different ECM components in the adipose stroma following macrophage depletion would provide additional insights into the mechanisms through which macrophages impact ECM homeostasis. It is important to note that pexidartinib, which was used to deplete macrophages in these studies through targeting CSF1R, inhibits additional kinases including c-Kit and, to a lesser extent, Flt3 (*DeNardo et al., 2011*). Thus, it is possible that in addition to macrophage depletion, inhibiting these pathways in other cell types may contribute to the observed stromal phenotypes. Our findings suggest that resident macrophages contribute to maintaining connective tissue homeostasis within the mammary gland, due in part to the regulation of ECM turnover. Nevertheless, further studies using additional methods of macrophage depletion, ideally using methods to delete Lyve-1[+] macrophages selectively, are required to confirm these findings.

In the normal mammary gland, HA is primarily found in the high molecular weight form, and results from in vitro studies have suggested a potential role for HA in epithelial branching (*Tolg et al., 2017*). We demonstrate here that HA is also associated with fibrous regions within the adipose stroma and the mammary capsule. Given the role for HA in the structural support of tissues and hydration, it seems likely that HA represents a key structural component of the mammary gland. Surprisingly, although we found that depletion of macrophages by CSF1R inhibition during ductal development led to enhanced ECM accumulation in the mammary gland, we did not observe significant defects in ductal elongation or branching. Previous studies have demonstrated that macrophage deletion in a *Csf1*-deficient mouse model, in which macrophages are not present during the initial stages of ductal development, results in reduced ductal elongation and branching during puberty (*Gouon-Evans et al., 2000*). Interestingly, targeting of CD11c[+] cells leads to enhanced ductal elongation in the mammary gland (*Plaks et al., 2015*). Furthermore, deletion of CD11b[+] cells, which reduced the stromal macrophage population, also enhanced ductal elongation (*Dawson et al., 2020*). The differences between these findings and our results may be the result of many factors, including differences in methods of macrophage depletion, timing of depletion and efficacy. Notably, CSF1R treatment only achieved 60% depletion efficiency in our studies, which may not be sufficient to impact ductal development significantly. Furthermore, we assessed mammary glands following two weeks of depletion, rather than one week as performed in the CD11b depletion model (*Dawson et al., 2020*), which may impact our ability to detect differences in elongation. Together, these findings suggest that myeloid regulation of ductal development in the mammary gland is complex and that identifying and modifying specific macrophage subpopulations may be required to understand fully the mechanistic functions of macrophages in mammary gland development and homeostasis. Furthermore, studies of stromal macrophages during processes that typically involve alterations in ECM, such as involution and aging (*Butcher et al., 2009*; *Li, 2005*; *Schedin and Keely, 2011*), may highlight additional functional consequences of altered ECM on epithelial morphology in the mammary gland.

Alterations in ECM accumulation in the breast are associated with changes in breast density (*Huo et al., 2015*). Furthermore, increased density is an established risk factor for developing breast cancer (*McCormack, 2006*). Our studies suggest a potential correlation between the presence of ECM-modulating macrophages and ECM accumulation. Unlike the mouse mammary gland, the density of stroma in the human breast varies widely, with regions of adipocytes interspersed with areas of connective fibers of varying densities. Furthermore, stromal density varies widely between individuals. Interestingly, areas of high mammographic density in the stroma are associated with reduced numbers of CD206[+] macrophages in human breast samples (*Huo et al., 2015*), consistent with the

hypothesis that CD206$^+$ macrophages impact ECM regulation in the normal mammary gland. Understanding what drives differences in macrophage localization in different regions of the mammary gland and in different individuals may provide insights into the mechanisms that contribute to breast density. Furthermore, understanding the mechanisms through which stromal composition is modulated in the normal mammary gland will ultimately lead to the development of strategies to modulate breast density and reduce breast cancer risk.

In addition to localization in the mammary gland, we also found that F4/80$^+$Lyve-1$^+$ cells accumulate in the peritumoral stroma surrounding mammary tumors. Given that the peritumoral regions of tumors are often associated with ECM modulation, this localization is consistent with the concept that these cells are involved in ECM regulation. The presence of Lyve-1$^+$ macrophages has been documented in tumors, and these cells have been implicated in lymphatic vessel remodeling and tumor progression (*Elder et al., 2018*; *Etzerodt et al., 2020*). Interestingly, Lyve-1$^+$ macrophages are lost during melanoma progression and melanoma tumor growth is enhanced in Lyve-1$^{-/-}$ mice , suggesting that Lyve-1-expressing cells have a tumor restraining role during tumor growth (*Dollt et al., 2017*). Further studies are required to differentiate between the functional roles of Lyve-1$^+$ macrophages and lymphatic endothelial cells in these models. By contrast, Lyve-1 expression is also associated with a specific macrophage subpopulation that was found to promote ovarian cancer progression and metastasis (*Etzerodt et al., 2020*). In addition, recent studies have also implicated macrophages expressing lymphatic-associated markers, including Lyve-1 and podoplanin, in mammary tumor metastasis through modulation of lymphatics (*Elder et al., 2018*; *Bieniasz-Krzywiec et al., 2019*). Although further studies are required to understand Lyve-1 macrophage function across different tumor types, these studies highlight the importance of identifying and understanding the functional contributions of distinct macrophage subpopulations in the context of both normal tissue homeostasis and tissue-specific disease.

# Materials and methods

## Key resources table

| Reagent type (species) or resource | Designation | Source or reference | Identifiers | Additional information |
|---|---|---|---|---|
| Strain, strain background (*Mus musculus*, female) | FVB/N | Envigo | FVB/NHsd | |
| Strain, strain background (*Mus musculus*, female) | Balb/c CD45.2 Mice | Jackson Laboratories | Stock No.: 000651 | Balb/cJ Mouse Strain |
| Strain, strain background (*Mus musculus*) | Balb/c CD45.1 Mice | Jackson Laboratories | Stock No.: 006584 | CByJ.SJL(B6)-*Ptprc*$^a$/J Mouse Strain |
| Antibody | violetFluor 450-Conjugated anti-CD45.1 (mouse monoclonal) | Tonbo Biosciences | Cat #: 75–0453 U025 RRID:AB_2621949 | Flow cytometry: 1:200 |
| Antibody | Alexa Fluor 700-Conjugated anti-CD45.2 (mouse monoclonal) | BioLegend | Cat #: 109822 RRID:AB_493731 | Flow cytometry: 1:200 |
| Antibody | BUV395-conjugated anti-Ly6G (rat monoclonal) | BioLegend | Cat #: 563978 RRID:AB_2716852 | Flow cytometry: 1:400 |
| Antibody | Brilliant Violet 421-Conjugated Anti-CD64 (mouse monoclonal) | BioLegend | Cat #: 139309 RRID:AB_2562694 | Flow cytometry: 1:100 |
| Antibody | Brilliant violet 711-conjugated anti-CD11c (Armenian hamster monoclonal) | BioLegend | Cat #: 117349 RRID:AB_2563905 | Flow cytometry: 1:200 |

*Continued on next page*

Continued

| Reagent type (species) or resource | Designation | Source or reference | Identifiers | Additional information |
|---|---|---|---|---|
| Antibody | PE/Cy7-conjugated anti-CD11b (rat monoclonal) | BioLegend | Cat #: 101216 RRID:AB_312799 | Flow cytometry: 1:200 |
| Antibody | PE-conjugated anti-F4/80 (rat monoclonal) | BioLegend | Cat #: 123109 RRID:AB_893498 | Flow cytometry: 1:200 |
| Antibody | Biotinylated anti-Lyve-1 (goat polyclonal) | R and D Systems | Cat #: BAF2125 RRID:AB_2138529 | Flow cytometry: 4 µg/mL |
| Antibody | Anti-CD16/CD32 (rat monoclonal) | eBioscience | Cat #: 14-0161-82 RRID:AB_467133 | Flow cytometry: 1:200 |
| Antibody | Rat anti-mouse monoclonal F4/80 (clone: CI: A3-1) | Bio-Rad laboratories | Cat# MCA497GA RRID:AB_323806 | IF 1:100 |
| Antibody | Goat polyclonal anti-mouse LYVE-1 (Ala24-Thr234) | R and D Systems | Cat# AF2125 RRID:AB_2297188 | IF 1:60 |
| Antibody | Anti-mannose receptor (rabbit polyclonal) | Abcam | Cat# ab64693 RRID:AB_1523910 | IF 1:1000 |
| Antibody | Anti-BrdU antibody [BU1/75 (ICR1)] | Abcam | Cat# ab6326 RRID:AB_305426 | IF 1:200 |
| Antibody | Hyaluronic acid binding protein, biotinylated | Millipore Sigma | Cat# 385911–50 UG | IF 1:100 |
| Antibody | Actin-smooth muscle (rabbit polyclonal) | Spring Bioscience | Cat# E2464 RRID:AB_95752 | IF 1:50 |
| Antibody | CD68 monoclonal antibody (514H12) | Thermo-Fisher Scientific | Cat# MA1-80133 RRID:AB_929283 | IF 1:100 |
| Antibody | Human LYVE-1 antibody (goat polyclonal) | R and D Systems | Cat# AF2089 RRID:AB_355144 | IF 1:60 |
| Antibody | Alexa 594 secondary antibody (donkey anti-mouse) | Thermo-Fisher Scientific | Cat# A-32744 RRID:AB_2762826 | IF 1:400 |
| Antibody | Alexa 594 secondary antibody (donkey anti-rat) | Thermo-Fisher Scientific | Cat# A-21209 RRID:AB_2535795 | IF 1:400 |
| Antibody | Alexa Fluor 488 conjugated streptavidin secondary antibody | Thermo-Fisher Scientific | Cat# S11223 | IF 1:400 |
| Antibody | Alexa 488 secondary antibody (donkey anti-goat) | Thermo-Fisher Scientific | Cat# A-11055 RRID:AB_2534102 | IF 1:400 |
| Antibody | Alexa 488 secondary antibody (goat anti-rabbit) | Thermo-Fisher Scientific | Cat# A-11008 RRID:AB_143165 | IF 1:400 |
| Antibody | Alexa 594 secondary antibody (goat anti-rat) | Thermo-Fisher Scientific | Cat # A-11007 RRID:AB_10561522 | IF 1:400 |
| Antibody | Alexa 594 secondary antibody (goat anti-rabbit) | Thermo-Fisher Scientific | Cat# A-11012 RRID:AB_2534079 | IF 1:400 |
| Antibody | Alexa Fluor 647 conjugated streptavidin secondary antibody | Thermo-Fisher Scientific | Cat# S-32357 | IF 1:400 |
| Commercial assay or kit | Hyaluronan enzyme-linked immunosorbent assay | Echelon Biosciences | Cat #: K-1200 | |
| Chemical compound, drug | Busulfan | Sigma-Aldrich | Cat #: B2635-10G | |

*Continued*

| Reagent type (species) or resource | Designation | Source or reference | Identifiers | Additional information |
|---|---|---|---|---|
| Chemical compound, drug | Pexidartinib | Selleck Chem | Cat #: S7818 | |
| Chemical compound, drug | Heparin | Sigma-Aldrich | Cat #: H3149-10KU | |
| Chemical compound, drug | Collagenase A | Sigma-Aldrich | Cat #: 10103586001 | |
| Chemical compound, drug | DNase I | Sigma-Aldrich | Cat #: DN25-100MG | |
| Software, algorithm | GraphPad Prism 8.0 | GraphPad Prism 8.0 | RRID:SCR_002798 | http://www.graphpad.com/scientific-software/prism/ |
| Software, algorithm | ImageJ 2.0.0 | ImageJ 2.0.0 | RRID:SCR_003070 | https://imagej.nih.gov/ij/download.html |
| Software, algorithm | Hisat2 | *Kim et al., 2015* | RRID:SCR_015530 | version 2.0.2 |
| Software, algorithm | DESeq2 software | *Love et al., 2014* | RRID:SCR_015687 | version 1.20.0 |
| Software, algorithm | ggplot2 | R | RRID:SCR_014601 | v3.0.0 |
| Software, algorithm | pheatmap | R | RRID:SCR_016418 | 1.0.12 |
| Other | Flow Cytometry Perm Buffer (10X) | Tonbo Biosciences | Cat #: TNB-1213-L150 | |
| Other | Fixable Viability Dye eFluor 780 | eBioscience | Cat #: 65-0865-14 | Flow cytometry: 1:1000 dilution |
| Other | APC-conjugated streptavidin | eBioscience | Cat #: 17-4317-82 | Flow cytometry: 1:100 dilution |
| Other | ProLong Gold Antifade Mountant (DAPI) | Thermo-Fisher Scientific | Cat# P36931 | |
| Other | Trichrome stain kit | Abcam | Cat# ab150686 | |
| Other | Antigen Retrieval Buffer (100X EDTA Buffer, pH 8.0) | Abcam | Cat# ab93680 | 1x |
| Other | Antigen unmasking solution, PH 6.0 | Thermo-Fisher Scientific | Cat# NC9401067 | 1x |

## Mice

Mice were purchased from Envigo Laboratories and Jackson Laboratories. All animal care and procedures were approved by the Institutional Animal Care and Use Committees of the University of Minnesota (#1909-37381A) and Tulane University (#710), and were in accordance with the procedures detailed in the Guide for Care and Use of Laboratory Animals.

## Human samples

The study was approved for exemption (#00008356) by the Institutional Review Board at the University of Minnesota and all patients had accepted the institutional standard consent for research utilization of clinical data and samples. All patient materials were de-identified following standard protocols. Tissue samples were derived from patients (n = 5) undergoing breast reduction (mammoplasty) surgery or cosmetic surgery. All patients' age was within the World Health Organization definition of reproductive age (15-49). Sections were reviewed and selected by an anatomic pathologist (ACN) to encompass representative areas of terminal duct lobular units with intralobular stroma and interlobular stroma with variable proportions of fibrous and adipose tissue. All patients were free of both benign and neoplastic breast disease.

## Immunostaining

For analysis of mouse mammary gland and tumor samples, mammary glands and tumors were harvested and fixed in 4% paraformaldehyde and paraffin embedded. 4T1-derived tumors were generated as previously described (*Irey et al., 2019*). Cells were authenticated by luciferase expression and tested negative for mycoplasma prior to injection into mammary fat pads as described (*Irey et al., 2019*). Following rehydration, sections were blocked for 1 hr in 10% normal donkey serum or normal goat serum and stained for F4/80 (1:100, BioRad #MCA497GA) and LYVE-1 (1:60, R and D Systems #AF2125) or CD206 (1:1000, Abcam # ab64693) at 4 ℃ overnight. Secondary antibody donkey anti-rat Alexa Fluor 594 (1:400, Thermo Fisher Scientific # A-21209) and donkey anti-goat Alexa Fluor 488 (1:400, Thermo Fisher Scientific # A-11055) or goat anti-rat Alexa Fluor 594 (1:400, Thermo Fisher Scientific # A-11007) and goat anti-rabbit Alexa Fluor 488 (1:400, Thermo Fisher Scientific # A-11008) were incubated for 1 hr at room temperature. The tissues were mounted with ProLong Gold Antifade DAPI (Thermo Fisher Scientific, #P36931). For BrdU staining, following dewaxing, rehydration, and heat-induced antigen retrieval in citric acid based buffer (1X, Thermo Fisher Scientific # NC9401067) at pH 6.0 for 20 min, tissues were incubated with BrdU (1:200, Abcam, #ab6326) at 4 ℃ overnight. Secondary antibody donkey anti-rat Alexa Fluor 594 (1:400, Thermo Fisher Scientific # A-21209) was incubated for 1 hr at room temperature. For HA staining, sections were incubated in HABP (1:100, Millipore Sigma #385911–50 UG) for 45 min at room temperature followed by incubation with secondary antibody Alexa Fluor 488 conjugated streptavidin (1:400, Thermo Fisher Scientific #S11223) or Alexa Fluor 647 conjugated streptavidin (1:400, Thermo Fisher Scientific # S-32357) for 1 hr. For analysis of human normal mammary tissues, heat-induced antigen retrieval was conducted in EDTA buffer (1X, Abcam #ab93680) at pH 8.0 for 20 min and staining was performed for CD68 (1:100, Thermo Fisher Scientific MA1-80133) and LYVE-1 (1:60, R and D Systems #AF2089) at 4 ℃ overnight. Secondary antibody donkey anti-mouse Alexa Fluor 594 (1:400, Thermo Fisher Scientific # A-32744) and donkey anti-goat Alexa Fluor 488 (1:400, Thermo Fisher Scientific # A-11055) were incubated for 1 hr at room temperature. For SMA staining, heat-induced antigen retrieval was performed in Tris-HCL buffer at pH 9.0 for 25 min. Tissues were stained for SMA (1:50, Spring Bioscience #E2464) at 4 ℃ overnight. Secondary antibody goat anti-rabbit Alexa Fluor 594 (1:400, Thermo Fisher Scientific # A-11012) was incubated for 1 hr at room temperature. Collagen trichrome staining was performed according to the manufacturer's protocol (Abcam, ab150686).

## Microscope image acquisition

Collagen trichrome staining and whole-mount images were taken on a Leica DM400B microscope at 10x objectives. Images were acquired using a Leica DFC310 FX camera and LAS V3.8 software, and were processed in imageJ. Images for all other immunostaining were captured using the Nikon C2+ confocal with the assistance of Mark Sanders at the University of Minnesota - University Imaging Centers. Specifically, for confocal imaging plus deconvolution: images were acquired in a Nikon Ni-E upright microscope equipped with a Plan Apo 10X, NA 0.45, Plan Apo 20x, NA 0.75 or Plan Apo 40x oil immersion objective lens, NA 1.30. Illumination was provided by 405, 488, 561 or 640 nm lasers fed through a single mode fiber to a Nikon C2+ confocal scan head. Emission filters used were 450/50, 525/50, 600/50 and 685/70 nm and the confocal aperture was set to the value of 30 μm. The images were subjected to automatic iterative deconvolution with Nikon Elements 5.20.

## In vivo macrophage depletion

The CSF1R inhibitor pexidartinib (Selleck Chemicals) was used to deplete macrophages. Pexidartinib was initially suspended and stored at 200 mg/mL in DMSO, and further suspended in 5% DMSO, 45% polyethylene glycol 3000 (Sigma-Aldrich), 5% Tween-80 (Sigma-Aldrich), and 45% ddH$_2$O for a working concentration of 10 mg/mL. FVB/N mice bred in-house at 5 weeks of age were given a working solution of pexidartinib at 45 mg/kg once daily by oral gavage for 2 weeks. Mammary glands were then either fixed in 4% paraformaldehyde for histology, processed for flow cytometry or lysed for HA analysis by ELISA as previously described (*Bohrer et al., 2014*). HA concentrations were normalized to mammary gland weight.

## Bone marrow chimeras

Female BALB/cJ mice (Jackson Stock No. 000651) at 6 weeks of age were exposed to 800 cGy of x-ray radiation using an RS-2000 X-ray irradiator (Rad Source) split evenly over two doses, 6 hr apart. Lead shields covering the #4 mammary glands were used to preserve any tissue-resident cells from exposure. To further deplete any cells that were still present after irradiation, mice were injected i.p. with 25 mg/kg Busulfan (Sigma-Aldrich) split evenly over two doses, the first one given 12 hr after initial irradiation and the second given 12 hr after that. Bone marrow was collected from the femurs and tibiae of female CD45.1 BALB/c mice (Jackson Stock No. 006584). Red blood cells were lysed, and $5*10^6$ cells were injected via the retroorbital vein into irradiated mice 24 hr after initial irradiation. After 6 weeks, blood was analyzed by flow cytometry to determine the level of chimerism.

## Tissue processing, flow cytometry and cell sorting

Blood monocytes were collected by facial vein puncture using 2U Heparin (Sigma-Aldrich) per 100 µL of blood as anticoagulant. After removal of inguinal lymph nodes, #4 mammary glands were dissected, minced following removal of lymph nodes, and digested with 1 mg/mL collagenase (Sigma-Aldrich) and 15 µg/mL DNase I (Sigma-Aldrich) for 45 min with shaking at 37℃. The digested mammary glands were filtered through a 70 µm cell strainer and pelleted by centrifugation at 500 g for 5 min. Red blood cells were lysed with buffer containing 150 mM ammonium chloride, 10 mM potassium bicarbonate, and 0.1 mM sodium EDTA at pH 7.4, and then resuspended in FACS buffer (PBS containing 2% FBS and 1 mM EDTA). Cells were stained for extracellular markers in FACS buffer at room temperature. The fluorochrome- and biotin-conjugated antibodies used were specific to mouse CD45.1 (Tonbo Biosciences, clone A20), CD45.2 (BioLegend, clone 104), Ly6G (BD Biosciences, Clone 1A8), CD64 (BioLegend, Clone X54-5/7.1), CD11c (BioLegend, Clone N418), CD11b (BioLegend, Clone M1/70), F4/80 (BioLegend, Clone BM8), Lyve-1 (R and D Systems, polyclonal), and streptavidin-APC (eBioscience). Fixable viability dye (eBioscience) was used to exclude dead cells. An antibody that is specific to mouse CD16 and CD32 (eBioscience, clone 93) was included to block Fc Receptor. Cells were fixed using 1% paraformaldehyde for 30 min at 4℃. Cells were permeabilized for 5 min at room temperature using 1X Flow Cytometry Perm Buffer (Tonbo biosciences). Intracellular staining was done using 1X Perm Buffer. A fluorochrome-conjugated antibody specific for CD206 (BioLegend, Clone C068C2) was used. CountBright Absolute Counting Beads (Life Technologies) were used for quantification of cell numbers. Flow cytometry was performed using an LSR Fortessa X-20 (BD Biosciences) and analyzed with FlowJo Software. Blood monocytes were identified as CD45+-Ly6G−CD11b+CD64+. Mammary-gland-resident macrophages were identified as CD45+-CD11c−CD11b+F4/80+. In bone marrow chimera experiments, host and donor cells were identified on the basis of the expression of CD45.2 or CD45.1, respectively. Cell sorting was performed with a BD FACSAria II cell sorter. For bulk RNA-seq data analysis, Lyve-1+ and Lyve-1− macrophages were identified and sorted as CD45+CD11b+F4/80+Lyve-1+ and CD45+CD11b+-F4/80+Lyve-1−, respectively. Mammary gland #4 from both sides of n = 4 mice at each timepoint (6 weeks and 10 weeks) were pooled for each sample. Samples were collected in biological duplicate. RNeasy Mini Kit (Qiagen) was used to extract RNA from sorted cells for RNA-sequencing.

## Bulk RNA sequencing

The Hisat2 (version 2.0.2) was used to map paired 50-bp reads to the mouse genome (mm10). The RSubreads (FeatureCounts) package was used for counting reads to genes (*Liao et al., 2013*). Genes of fewer than 300-bp were removed. PCA revealed a clear outlier in one of the 6-week samples that had severe sequencing problems (only 2 million reads mapped to genes and the most highly expressed genes were a group of RNA genes, suggesting that mRNA enrichment during library prep failed in this sample). This sequencing outlier was removed from any further analysis.

Differential gene expression analysis was performed using the DESeq2 software (version 1.20.0) (*Love et al., 2014*). Plots were created in R with ggplot2 (v3.0.0) and pheatmap (1.0.12) packages.

### Immune cell enrichment (scRNA-seq)

Mammary glands were harvested from 10-week-old diestrus FVB/NJ mice (n = 3). After removing the lymph nodes, the thoracic (#3) and inguinal (#4) glands were pooled, minced and incubated in DMEM containing 2 mg/ml collagenase A (Roche) and two units/ml DNase (Sigma) at 37℃ for 12 min, and single cells were purified as previously described (*Carron et al., 2017*). After red blood cell depletion with ACK lysis buffer (Thermo Fisher Scientific), single cells were incubated with mouse CD45 microbeads (Miltenyi Biotec) and CD45$^+$ cells were enriched according to the manufacturer's protocol and prepared for scRNA-seq.

### Single-cell RNA sequencing

Thirteen thousand individual cells with a viability of >88% were targeted for GEM generation and barcoding using 10x GemCodeTechnology, which allows the partitioning thousands of cells into nanoliter-scale Gel Bead-In-Emulsions (GEMs), applying ~750,000 barcodes to index the transcriptome of each cell separately. Full-length barcoded cDNA was generated and amplified by PCR, followed by enzymatic fragmentation, end-repair, A-tailing, and adaptor ligation. Single-cell libraries were run using paired-end sequencing with single indexing with the NextSeq 550 platform. The 10x genomic cellranger pipeline (version 3.1.0) was used to demultiplex, map (mm10 genome) and generate counts for the single-cell sequence. The Seurat R Package (version 3.1.1) was used to analyze the single-cell data. Data were filtered to include cells containing 200 to 5000 unique gene counts and the expression of at least 200 genes by five cells. Global-scaling normalization was applied using defaults. Highly variable features (2000 genes) were identified using the vst selection method. The data were then scaled to regress out sequencing depth, and linear dimensional reduction (PCA) was performed on the most variable genes. The first 20 principle components were used to construct a Shared Nearest Neighbor Graph, and clustering was performed using the Louvain algorithm with different resolution parameters. To visualize the clusters, non-linear dimensional reduction (UMAP) was performed. Differential gene expression (DE) analysis using the 'wilcox' method was performed two different ways. To define the accurate resolution of clusters, DE was performed between a cluster and all other clusters at different resolutions. Once clusters were defined using a specific resolution, DE was performed between specific clusters.

### Pathway analysis of DE genes from both bulk and single-cell analysis

Ranked lists were created from DE genes lists by taking the negative $\log_{10}$ of the p-value and adding the direction of change from the fold-change. This results in a ranked list that is arranged by the most significant p-value. Gene set enrichment analysis (GSEA) (*Subramanian et al., 2005*) was performed using the R-based clusterProfiler package (version 3.8.1) (*Yu et al., 2012*) with the Molecular Signature Databases (MsigDB v7) C5. To check whether the Lyve-1 positive clusters in the single-cell RNA sequencing samples were enriched for genes in the bulk RNA sequencing DE results, gene sets were created (adjusted p-value <0.1 and a fold-change >0.25) to use for enrichment (shown in *Figure 2F*).

### Statistical analysis

Experiments were performed at least three times. Statistical tests including the Welch's t-test, the Mann-Whitney U test, and the two-way ANOVA test were used as indicated in the Figure Legends. Means ± standard errors are presented unless otherwise stated. p-values <0.05 were considered statistically significant.

## Acknowledgements

The authors thank Dr Yi Shan (Xuanwu Hospital, Capital Medical University) for generating the mouse schematic and Ms Bobbie Daughters for her invaluable work to secure the IRB approval and to acquire the human tissue samples used in this study. The authors would also like to acknowledge the technical help provided by the University of Minnesota Imaging Core (UIC), University of Minnesota Genomics Core and Dr Kejing Song at the Tulane Center for Translational Research in Infection and Immunity NextGen Sequencing Core. This work was supported by a T32 fellowship (OD010993) to PMW; American Cancer Society Post-doctoral Fellowship (#132570-PF-18-140-01-CSM) to DNH;

American Cancer Society Clinical Scholar Development Grant (#132574-CSDG-18-139-01-CSM) to ACN; NIH R01CA212518 to HLM; and NIH funding R01CA215052, R01HD095858 and R21CA235385 to KLS.

## Additional information

### Funding

| Funder | Grant reference number | Author |
|---|---|---|
| National Institutes of Health | T 32 fellowship, OD010993 | Patrice M Witschen |
| American Cancer Society | Post-doctoral fellowship, 132570-PF-18-140-01-CSM | Danielle N Huggins |
| American Cancer Society | Clinical Scholar Development Grant, 132574-CSDG-18-139-01-CSM | Andrew C Nelson |
| National Institutes of Health | R01CA212518 | Heather L Machado |
| National Institutes of Health | R01HD095858 | Kathryn L Schwertfeger |
| National Institutes of Health | R01CA235385 | Kathryn L Schwertfeger |
| National Institutes of Health | R01CA215052 | Kathryn L Schwertfeger |

The funders had no role in study design, data collection and interpretation, or the decision to submit the work for publication.

### Author contributions

Ying Wang, Thomas S Chaffee, Data curation, Formal analysis, Investigation, Methodology, Writing - review and editing; Rebecca S LaRue, Formal analysis, Writing - review and editing; Danielle N Huggins, Supervision, Methodology, Writing - review and editing; Patrice M Witschen, Ayman M Ibrahim, Data curation, Methodology; Andrew C Nelson, Conceptualization, Supervision, Validation, Writing - review and editing; Heather L Machado, Conceptualization, Supervision, Funding acquisition, Writing - review and editing; Kathryn L Schwertfeger, Conceptualization, Supervision, Funding acquisition, Writing - original draft

### Author ORCIDs

Ying Wang https://orcid.org/0000-0001-9052-0023
Thomas S Chaffee https://orcid.org/0000-0001-5535-1535
Kathryn L Schwertfeger https://orcid.org/0000-0002-9755-7774

### Ethics

Human subjects: The study was approved for exemption (#00008356) by the Institutional Review Board at the University of Minnesota and all patients had accepted the institutional standard consent for research utilization of clinical data and samples. All patient materials were de-identified following standard protocols.

Animal experimentation: All animal care and procedures were approved by the Institutional Animal Care and Use Committees of the University of Minnesota (protocol #1909-37381A) and Tulane University (protocol #710) and were in accordance with the procedures detailed in the Guide for Care and Use of Laboratory Animals.

### Decision letter and Author response

Decision letter https://doi.org/10.7554/eLife.57438.sa1
Author response https://doi.org/10.7554/eLife.57438.sa2

# Additional files

## Supplementary files

• Supplementary file 1. List of 155 differentially expressed genes in CD45$^+$CD11b$^+$F4/80$^+$Lyve-1$^-$ and CD45$^+$CD11b$^+$F4/80$^+$Lyve-1$^+$ bulk RNA-seq analysis.

• Supplementary file 2. List of the top 100 differentially regulated genes in each cluster from the scRNA-seq analysis.

• Supplementary file 3. GSEA analysis using gene sets created from a differentially expressed ranked list from the bulk RNA-seq data (adjusted p-value <0.1 and a fold-change >0.25) to determine enrichment of cluster 4.

• Supplementary file 4. List of top gene sets identified by GSEA analysis including core enriched genes.

• Transparent reporting form

## Data availability

Sequencing data have been deposited in GEO under accession codes GSE148207 and GSE148209.

The following datasets were generated:

| Author(s) | Year | Dataset title | Dataset URL | Database and Identifier |
|---|---|---|---|---|
| Wang Y, Chaffee TS, LaRue RS, Huggins DN, Witschen PM, Ibrahim AM, Nelson AC, Machado HL, Schwertfeger KL | 2020 | Resident macrophages promote homeostasis of hyaluronan-associated extracellular matrix in the mammary gland stroma | https://www.ncbi.nlm.nih.gov/geo/query/acc.cgi?acc=GSE148207 | NCBI Gene Expression Omnibus, GSE148207 |
| Machado HL, LaRue RS, Schwertfeger KL | 2020 | Single cell analysis of immune cell populations in mammary glands from 10-week-old mice | https://www.ncbi.nlm.nih.gov/geo/query/acc.cgi?acc=GSE148209 | NCBI Gene Expression Omnibus, GSE148209 |

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
