## [Decision Letter]

**Acceptance summary:**

Macrophages are immune cells that not only respond to infection but also play important roles in development and maintenance of tissues. While these cells have been implicated in the development of the mammary gland as well as in mammary involution after pregnancy, this paper reports the function of a population of macrophages in the homeostatic maintenance of the virgin mammary gland.

**Decision letter after peer review:**

Thank you for submitting your article "Resident macrophages promote homeostasis of hyaluronan-associated extracellular matrix in the mammary gland stroma" for consideration by *eLife*. Your article has been reviewed by two peer reviewers, and the evaluation has been overseen by a Reviewing Editor and Satyajit Rath as the Senior Editor. The following individual involved in review of your submission has agreed to reveal their identity: F.M. Davis (Reviewer #2).

The reviewers have discussed the reviews with one another and the Reviewing Editor has drafted this decision to help you prepare a revised submission.

Summary:

This study by Wang, Chaffee and colleagues characterizes a population of tissue-resident macrophages enriched in the stroma of the mammary gland. The authors show that LYVE-1+ cells (~35% of macrophages in the non-pregnant mouse gland) are abundant in the mammary adipose and capsule, with little presence in/around epithelial structures. They confirm these findings in non-pregnant, normal human gland. The authors also employ a series of gene expression, single cell sequencing and immunostaining approaches to further characterize this population. In aiming to define their function, the study shows that pharmacologically-mediated macrophage depletion alters the mammary ECM. Overall this is an important area of research. The experiments are well-conducted and described and include appropriate controls. However, the conclusions should more carefully reflect the findings. Specifically, in the absence of selective ablation of LYVE-1+ cells or LYVE-1 expression, it cannot be concluded at this point that this macrophage population serves to maintain HA and collagen levels.

Revisions:

In the absence of selective depletion of LYVE-1+ macrophages and genetic ablation of LYVE-1 in these cells, it cannot be concluded that LYVE-1 expression on the LYVE-1+ macrophage population in the mammary gland serves to maintain HA and collagen levels. If the authors have data to support this conclusion, it would be important to include such data in the revision of this manuscript. In absence of this type of data, the authors should revise the claimed mechanism and indicate in the Discussion that future studies, including approaches described above, are needed to unambiguously conclude on the requirement of LYVE-1 for the maintenance of HA and collagen levels.

Another important concern relates to the quantification of ECM components. Can the authors include an additional assay beyond immunostaining intensity?

---

## [Author Response]

Revisions:In the absence of selective depletion of LYVE-1+ macrophages and genetic ablation of LYVE-1 in these cells, it cannot be concluded that LYVE-1 expression on the LYVE-1+ macrophage population in the mammary gland serves to maintain HA and collagen levels. If the authors have data to support this conclusion, it would be important to include such data in the revision of this manuscript. In absence of this type of data, the authors should revise the claimed mechanism and indicate in the Discussion that future studies, including approaches described above, are needed to unambiguously conclude on the requirement of LYVE-1 for the maintenance of HA and collagen levels.

We fully agree that we are unable to make conclusions regarding the specific functions of LYVE1 expression in the macrophages. We have tempered the wording throughout the manuscript (Abstract, Introduction, last paragraph,, Discussion, first paragraph) and have added additional text to the Discussion to indicate that future studies using selective ablation/deletion of LYVE-1 in this population are required to make this conclusion (subsection “Lyve-1^+^ cells are associated with hyaluronan-enriched regions in the mammary gland and in mammary tumors”, second paragraph, Discussion, fourth paragraph).

Another important concern relates to the quantification of ECM components. Can the authors include an additional assay beyond immunostaining intensity?

We have included additional quantification of HA using an HA ELISA assay (Figure 6D) that supports the findings from the immunostaining analysis (subsection “Macrophage depletion impacts ECM in the stroma”).